# Are Vision Language Models Ready for Clinical Diagnosis? A 3D CT Benchmark for Lesion-centric Visual Question Answering

## Abstract

Vision-Language Models (VLMs) have shown promise in various 2D visual tasks, yet their readiness for 3D clinical diagnosis remains unclear due to stringent demands for recognition precision, reasoning ability, and domain knowledge. To systematically evaluate these dimensions, we present CTLesionVQA, a diagnostic visual question answering (VQA) benchmark targeting abdominal lesions in CT scans. It comprises 9,262 CT volumes (3.7M slices) from 17 public datasets, with 395K expert-level questions spanning four categories: *Recognition*, *Measurement*, *Visual Reasoning*, and *Medical Reasoning*. CTLesionVQA introduces unique challenges, including small tumor detection and clinical reasoning across 3D anatomy. Benchmarking four advanced VLMs (RadFM, M3D, Merlin, CT-CHAT), we find current models perform adequately on measurement tasks but struggle with lesion recognition and reasoning, and do not meet clinical needs. Two key insights emerge: (1) large-scale multimodal pretraining plays a crucial role in a VLM's performance, making RadFM stand out among all VLMs. (2) Our dataset exposes critical differences in VLM components, where proper image preprocessing and design of vision modules significantly affect 3D perception.

## 1 Introduction

Vision-language models (VLMs) (Zhang et al., 2024a) have achieved impressive performance across general visual reasoning tasks. However, applying them to medical imaging introduces significantly more stringent requirements, due to the high-stakes nature of clinical decision-making. Existing medical VLMs (Wu et al., 2023; Hamamci et al., 2024b; Bai et al., 2024; Blankemeier et al., 2024) have typically been evaluated on simplified or exploratory benchmarks that do not reflect real-world clinical complexity. This raises a critical question: *Are 3D medical VLMs precise and intelligent enough for clinical diagnosis?* Clinical diagnosis refers to the judgment about the nature of a patient's disease, made by imaging studies in the context of this work. To address this, there is a pressing need for a high-quality and diagnostically meaningful benchmark that enables rigorous evaluation of state-of-the-art (SOTA) models in realistic clinical contexts.

A number of medical VQA benchmarks (Lin et al., 2023; Hartsock & Rasool, 2024) have been proposed to evaluate the capabilities of VLMs. However, they suffer from five limitations that hinder their utility as standardized benchmarks: **First**, *limited scale and diversity*. Due to the high cost and time required for expert annotation, most clinical datasets remain small in scale and lack diversity (*e.g.*, VQA-Rad (Lau et al., 2018), VQA-Med (Ben Abacha et al., 2021), Open-I (Demner-Fushman et al., 2016), EndoVis 2017 (Allan et al., 2019)). **Second**, *reliance on 2D and web-sourced images*. Many recent large-scale datasets, including SLAKE (Liu et al., 2021), PMC-VQA (Zhang et al., 2023b), OmniMedVQA (Hu et al., 2024), and PathVQA (He et al., 2020), are constructed using 2D images from public websites or scientific publications. They do not adequately reflect the 3D volumetric nature of clinical imaging. **Third**, *lack of consistent and reliable evaluation metrics*. Automated metrics such as BLEU and ROUGE are not well-suited for evaluating short, factual medical answers, as they often fail to capture semantic correctness (Lin et al., 2023). While human evaluation (Kovaleva et al., 2020) aligns more closely with clinical judgment, it is costly and difficult to reproduce. **Fourth**, *oversimplified questions*. Existing datasets often include experimental or toy questions (*e.g.*, organ, phase, or plane recognition (Bai et al., 2024)). However, real-world

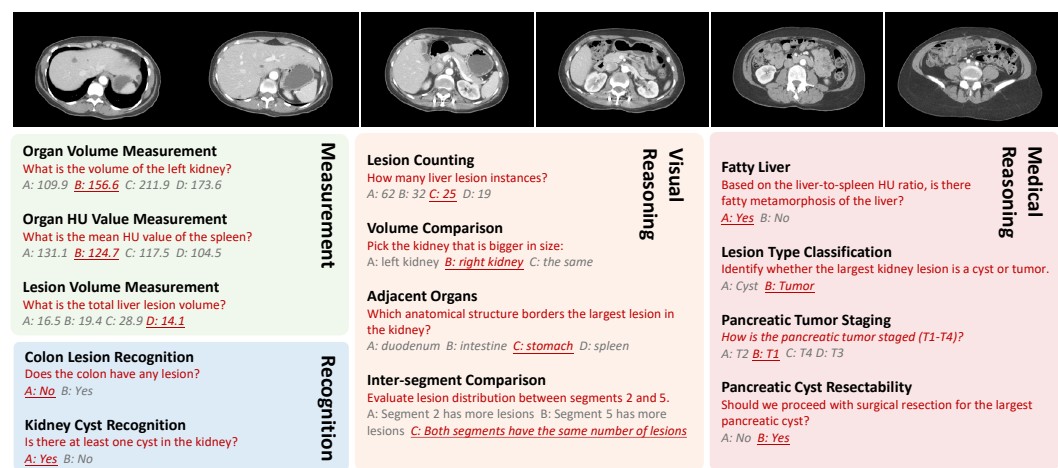

Figure 1: Overview of tasks in the CTLesionVQA benchmark. The dataset covers four core clinical question types, totaling 29 subtypes. Tasks include numerical quantification (*e.g.*, organ volume, Hounsfield Unit (HU) value), lesion recognition, spatial reasoning (*e.g.*, comparisons, adjacency), and high-level clinical diagnosis (*e.g.*, tumor staging, resectability). Each question is paired with image evidence and formatted for either multiple-choice or free-text answer prediction, enabling evaluation of both perceptual and diagnostic reasoning in VLMs.

clinical questions frequently require measurement and reasoning with anatomical knowledge and clinical context. **Fifth**, *limited accessibility*. Some datasets are based on private institutional data, which restricts broad usage and reproducibility in the research community. To date, no existing medical VQA dataset integrates large-scale, multi-source 3D imaging data with high-quality expert annotations and clinically structured question hierarchies into a unified and accessible benchmark.

To bridge these gaps, we introduce **CTLesionVQA** (Fig. 1), a comprehensive dataset for evaluating VLMs in abdominal CT-based clinical diagnostics. CTLesionVQA comprises 9,262 CT volumes (3.7M slices) derived from 17 public datasets and 88 centers. 23 radiologists participated in the annotation and all questions are generated using templates from patterns found in structured radiology reports and medical literature, ensuring clinical relevance. CTLesionVQA comprises 395K question-answer pairs covering four hierarchical diagnostic tasks: *Recognition*, *Measurement*, *Visual Reasoning*, and *Medical Reasoning*. The former two types require models to precisely perceive organs and lesions. Built upon them, the latter two require models to intelligently reason about anatomical structures and apply external medical knowledge. The dataset mirrors the diagnostic reasoning hierarchy used by radiologists.

Through extensive benchmarking experiments using four SOTA VLMs—RadFM (Wu et al., 2023), M3D (Bai et al., 2024), Merlin (Blankemeier et al., 2024), and CT-CHAT (Hamamci et al., 2024a)—we provide detailed analyses that expose fundamental strengths and weaknesses of existing approaches. The results show that SOTA VLMs are better at large objects like organs, but struggle significantly with identifying small lesions and performing reasoning tasks that involve them. Our in-depth analysis also reveals the impact of basic visual tasks on the reasoning tasks, as well as the relationship between lesion characteristics and recognition performance.

Our contributions are summarized as follows: (1) We release CTLesionVQA, the first large-scale 3D VQA benchmark for lesion diagnosis with expert annotations and question hierarchies. (2) We present a comprehensive empirical analysis of VLMs, revealing key challenges in lesion recognition and reasoning. (3) We provide open-source data, code, and tools, and commit to maintaining the benchmark via recurring challenges.

## 2 RELATED WORK

**Medical Visual Question Answering.** VQA has become an important benchmark task for evaluating multimodal clinical systems. Early medical VQA datasets such as VQA-RAD (Lau et al., 2018)

and VQA-Med 2018–2020 (Ben Abacha et al., 2021) featured small-scale 2D image collections with a limited range of question types, often relying on templates or handcrafted QA pairs. Subsequently, more diverse datasets like PathVQA (He et al., 2020), SLAKE (Liu et al., 2021), and RadVisDial (Kovaleva et al., 2020) introduced pathology slides, structured medical knowledge, and dialog-style multi-turn QA, broadening the scope beyond simple abnormality detection. Recently, larger-scale benchmarks such as PMC-VQA (Zhang et al., 2023b), and OmniMedVQA (Hu et al., 2024) incorporate richer question types, hierarchical QA structures, and answers grounded in dense clinical reports. These datasets have shifted the field's emphasis from classification to explanation, reasoning, and domain adaptation. Notably, as public datasets like RadGenome Chest-CT (Zhang et al., 2024b), RadGenome Brain-MRI (Lei et al., 2024), and AMOS (Ji et al., 2022) have expanded, the feasibility of 3D medical VQA has improved significantly, enabling the creation of volumetric benchmarks requiring spatial reasoning and multi-slice integration (Hamamci et al., 2024a). This transition from static 2D diagnosis to rich, multi-view 3D reasoning reflects the evolution of the task's complexity and its alignment with real-world clinical scenarios.

**Medical Vision-Language Models.** VLMs designed for medical imaging tasks have undergone significant architectural and methodological evolution. Earlier systems largely used ResNet-based (He et al., 2016) image encoders paired with LSTM or Transformer-based text encoders (Sharma et al., 2021; Abacha et al., 2018; Peng et al., 2018; Ren & Zhou, 2020). Recent models have transitioned to Vision Transformer (ViT) backbones (Hamamci et al., 2024c), which better preserve spatial and contextual information, and allow for more expressive visual representations. Pretraining objectives have shifted from contrastive learning (CLIP-style) (Wang et al., 2022; Zhang et al., 2023a) to encoder-decoder paradigms, where image features are passed into large language decoders for autoregressive medical text generation (Chen et al., 2025). Concurrently, models like Med-PaLM (Tu et al., 2024), LLaVA-Med (Li et al., 2023), Med-Gemini (Yang et al., 2024b), and RadFM (Wu et al., 2023) began to scale both in terms of language model size and the diversity of medical tasks they support. Another recent trend is the support for 3D inputs, where ResNets/ViTs are adapted to volumetric data (Bai et al., 2024; Blankemeier et al., 2024) and 3D image-text pretraining. Additionally, the pretraining corpora have evolved to include multiple clinical data sources—reports, textbooks, biomedical QA pairs, and PACS metadata—making modern medical VLMs increasingly robust and generalizable.

# 3 CTLESIONVQA DATASET

## 3.1 OVERVIEW

The design of **CTLesionVQA** is inspired by the compositional reasoning framework in CLEVR (Johnson et al., 2017), adapted to the clinical context of diagnostic decision-making in abdominal CT. Our goal is to build a dataset that reflects real-world diagnostic needs while exposing the performance boundaries of VLMs under varying levels of task complexity. CTLesionVQA comprises basic and compositional question types, ranging from simple recognition and measurement to sophisticated visual and clinical reasoning, thus enabling detailed analysis of VLM behavior and limitations.

To overcome the limitations highlighted in Section 1, we construct a large-scale benchmark featuring: (1) **High data volume and diversity**: We curate 3D CT scans from 17 public datasets, encompassing over 9,000 volumes and millions of slices. (2) **Volumetric 3D supervision**: Unlike most prior benchmarks limited to 2D images, our dataset operates on full CT volumes, aligning with clinical diagnostic practice. (3) **Standardized evaluation metrics**: To ensure reproducibility and clinical relevance, we use task-specific metrics: accuracy for multiple-choice questions, exact match for free-text categorical answers, and mean relative accuracy (MRA) (Yang et al., 2024a) for quantitative numerical prediction. (4) **Clinical question design**: These question types align with key steps in radiological workflows, where clinicians must not only perceive features but also reason about their diagnostic significance. Importantly, reasoning questions are systematically constructed by composing functions over outputs from the recognition and measurement stages. This can ensure a dependency structure among questions, acting as a smart way to enforce multi-step reasoning.

The dataset contains 355,962 training QA pairs from 8,334 CT and 39,650 testing QA pairs from 928 CT. Its statistics of tasks and CT samples are shown in Fig. 2.

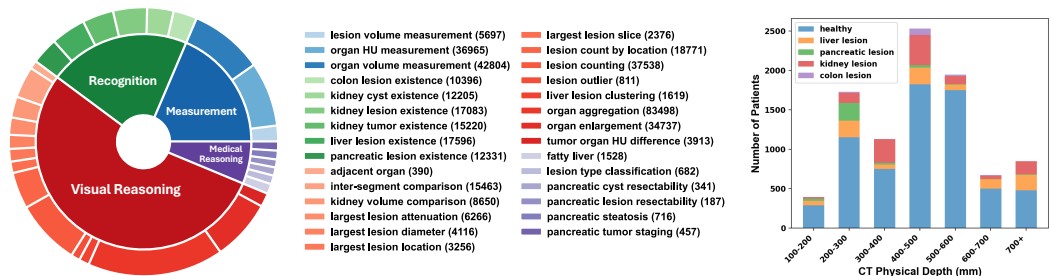

Figure 2: Statistics of CTLesionVQA. Left: the distribution of QA pairs for tasks across four main types. Right: distribution of CT volumes *w.r.t.* CT physical depth (z-axis) and patient types.

Table 1: Overview of public abdominal CT datasets that are collected in CTLesionVQA. Our reported number of CT volumes may differ from the original publications, as some CT volumes are reserved for further validation purposes. The number of CT volumes in CTLesionVQA is lower than the sum of datasets 1–17 due to the removal of duplicated samples.

| dataset (year) [source] | # of volumes | # of centers | dataset (year) [source] | # of volumes | # of centers |
|---|---|---|---|---|---|
| 1. CHAOS (2018) [link] | 20 | 1 | 2. Pancreas-CT (2015) [link] | 42 | 1 |
| 3. BTCV (2015) [link] | 47 | 1 | 4. LiTS (2019) [link] | 131 | 7 |
| 5. CT-ORG (2020) [link] | 140 | 8 | 6. WORD (2021) [link] | 120 | 1 |
| 7. AMOS22 (2022) [link] | 200 | 2 | 8. KiTS (2020) [link] | 489 | 1 |
| 9–14. MSD CT Tasks (2021) [link] | 945 | 1 | 15. AbdomenCT-1K (2021) [link] | 1,050 | 12 |
| 16. FLARE'23 (2022) [link] | 4,100 | 30 | 17. Trauma Detect. (2023) [link] | 4,711 | 23 |

## 3.2 DATASET CONSTRUCTION

**Data Collection.** We compile 9,262 abdominal CT volumes from 17 public datasets (Tab. 1), encompassing diverse acquisition protocols, scanners, and patient populations from 88 centers to ensure robust coverage of organs and pathologies. To provide high-quality annotations, 23 radiologists, including 7 senior specialists, 11 board-certified general radiologists, and 5 residents under specialist supervision (detailed in Appendix C) annotated all abdominal organs and 7,629 lesions over six months. They include 3,067 liver, 4,078 kidney, 351 pancreatic, and 131 colon lesions. Kidney tumors and cysts were labeled when distinguishable; ambiguous cases were marked as non-specific lesions. All annotations were performed as 3D segmentation masks and double-checked for consensus. The masks' quality is evaluated in Appendix D.

**Question Generation.** Inspired by the CLEVR dataset's methodology (Johnson et al., 2017), our question generation pipeline (Figure 3) uses a series of functional programs to create complex diagnostic questions from basic anatomical and pathological data. Each functional program can be thought of as a command or a query that is executed on the structured metadata extracted from the CT scans. For instance, a simple program might retrieve the volume of a specific organ, while a more complex one could involve multiple steps, such as identifying all tumors in the liver, counting their numbers in each liver segment, and comparing their spatial distribution against a predefined clinical threshold. This modular approach allows for the systematic generation of a wide array of clinically relevant questions with verifiable answers, ensuring the dataset's quality and utility for evaluating the reasoning capabilities of VLMs. A detailed breakdown of the metadata extraction and the full set of functional programs used for question generation can be found in Appendices A and B, respectively. In this work, we define **clinical diagnosis** as the process of interpreting radiological images to identify and assess the clinical implications of lesions, especially tumors. Based on this, we construct four types of diagnostic tasks with increasing complexity:

**Measurement (3 subtypes):** Numerical assessments like organ volume and HU value.

**Recognition (6 subtypes):** Recognize lesions like tumors and cysts.

**Visual Reasoning (14 subtypes):** Compositional logic-based tasks including spatial comparisons (*e.g.*, "Which segment contains more lesions?"), counting and localization of lesions, and lesion-organ relationship (*e.g.*, "Are there adjacent organs for a specific tumor?").

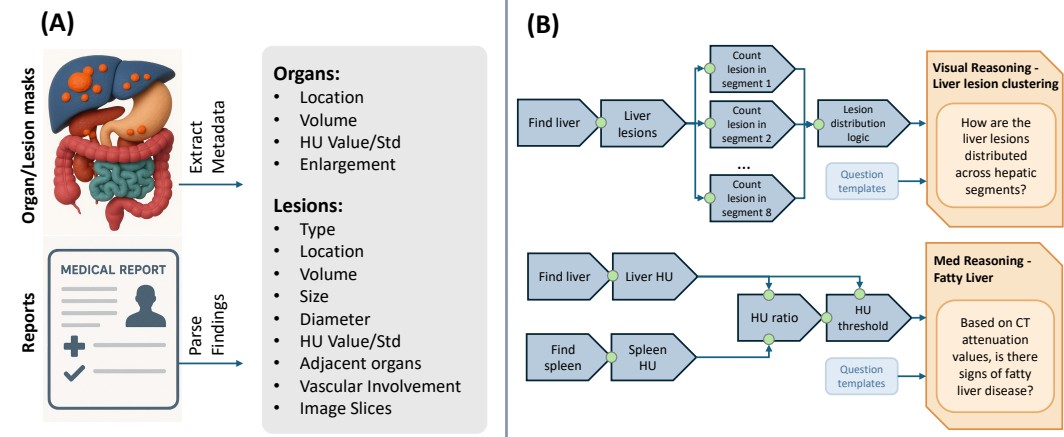

Figure 3: Question generation in the CTLesionVQA dataset. **(A)** Structured metadata is extracted from organ and lesion segmentation masks (*e.g.*, location, volume, HU value, enlargement) and parsed radiology reports (*e.g.*, lesion type, adjacent organs, vascular involvement). **(B)** These metadata are used to define modular logic programs for different question types. Each program maps to one of four task types and 29 subtypes, and is rendered into natural language using predefined templates.

**Medical Reasoning (6 subtypes):** This category includes clinical inference tasks that require external knowledge from established medical literature and guidelines, fatty liver diagnosis (Zeb et al., 2012), kidney lesion diagnosis (Agochukwu et al., 2017), pancreatic steatosis diagnosis (Guneyli et al., 2022), pancreatic cyst resectability (Johns Hopkins Medicine, 2022), and pancreatic tumor staging (Bassi et al., 2025). These tasks are intentionally designed to represent foundational, 1-2 step diagnostic inferences that are programmatically verifiable and grounded in quantitative evidence from the image (*e.g.*, HU-ratio-based diagnosis). They serve as a litmus test for a VLM's ability to connect visual findings with essential medical knowledge, which is a prerequisite for more complex problems.

Once we get a ground truth answer for a question, error choices for multi-choice questions are sampled from the empirical distributions of organ volumes, HU values, lesion counts, and categorical labels across the dataset. For example, volume-related distractors are selected by perturbing the ground-truth value within clinically plausible bounds, while maintaining class balance and avoiding implausible answers. For categorical questions, distractors are drawn from frequently co-occurring values in the same task. In addition, we hide the choices to serve as free-text questions in the CTLesionVQA dataset. In this case, the model must predict text-form answers. To mitigate the limited linguistic diversity of rule-based templates, we use 10 templates per question subtype with varying phrasing and entity ordering to mitigate this problem.

## 4    EVALUATION ON CTLESIONVQA BENCHMARK

### 4.1    DETAILS OF VLM EVALUATION

To evaluate the capabilities of current VLMs in solving volumetric medical VQA tasks, we benchmark four representative models: RadFM (Wu et al., 2023), M3D (Bai et al., 2024), Merlin (Blankemeier et al., 2024), and CT-CHAT (Hamamci et al., 2024a). Each model adopts a different architectural design and training strategy to integrate 3D visual information with language modeling, as summarized in Tab. 2. Each model varies in its architectural design and training procedure. RadFM leverages a pre-trained LLaMA2-13B (Touvron et al., 2023) with a perceiver resampler (Alayrac et al., 2022) to fuse 3D features, and fine-tunes both the LLM decoder and projector. M3D supports two LLM backbones (LLaMA2 and Phi-3 (Abdin et al., 2024)) and adopts a spatial pooling perceiver module to aggregate 3D volume features. In contrast, Merlin uses a simpler architecture with a ResNet-based visual encoder (Hara et al., 2018) and a single-layer linear projector. Finally, CT-CHAT adopts a ViT-based encoder tailored for CT (Hamamci et al., 2024c) and employs a CoCa attentional pooling (Yu et al., 2022). All the latter three models are fine-tuned with LoRA. We summarize the training hyperparameters and computation costs in Appendix E.

Table 2: Model architectures and training settings for four benchmarked VLMs. We use the original code base of the methods and follow their training hyperparameters for VQA tasks.

| Component | RadFM | M3D (LLaMA2 / Phi-3) | Merlin | CT-CHAT |
|---|---|---|---|---|
| Vision Encoder | 3D ViT | 3D ViT | 3D ResNet | CT ViT |
| Input image size | [256,256,64] | [256,256,32] | [224,224,160] | [300,300,600] |
| 3D CT spacing | direct resize | direct resize | [1.5mm, 1.5mm, 3mm] | [1.5mm, 1.5mm, 1.5mm] |
| LLM Decoder | LLaMA2-13B | LLaMA2-7B / Phi-3-4B | RadLLaMA-7B | LLaMA3.1-7B |
| Projector | Perceiver Resampler | 3D Pooling + 2-layer MLP | 1-layer FC | CoCa Attentional Pooling |
| Visual tokens/image | 32 | 256 | 1 | 256 |
| Pretraining Data | 16M 2D+3D multimodal | 120K 3D CT | 14K 3D Abdomen CT | 50K 3D Lung CT |
| LLM tuning | full | LoRA (r=16) | LoRA (r=128) | LoRA (r=128) |
| Projector tuning | ✓ | ✓ | ✓ | ✓ |
| Vision tuning | ✗ | ✗ | ✗ | ✗ |
| Learning rate | 5e-6 | 5e-5 | 1e-4 | 2e-5 |

## 4.2 ANALYSIS OF BENCHMARKING RESULTS

We analyze model performance (Table 3) across three axes: task format, diagnostic competence, and model architecture. Our findings are statistically robust, with narrow confidence intervals reported in Appendix F.

**Task Format:** Four of five models perform better in the multi-choice setting, where candidate options provide inductive constraints. For example, in *lesion counting*, models generate plausible answers with choices, but default to zero in free-text, indicating weak numeracy, especially for small structures. However, this advantage diminishes for Yes/No-style questions in *recognition* and binary reasoning tasks (*e.g.*, *fatty liver*, *pancreatic steatosis*), where free-text matches or even slightly exceeds multi-choice. This may stem from pretraining on open-ended generation, which favors categorical outputs.

**Diagnostic Readiness:** Model competencies are highly uneven across diagnostic tasks. They demonstrate promise in basic and recognition-heavy tasks, their applicability to real-world diagnostics is currently limited by weak visual signal, unreliable numeracy, and shallow reasoning chains.

a. *Measurement tasks* are the most tractable, with all models significantly outperforming random-guess, likely due to the high signal-to-noise ratio of large targets like organs.

b. *Recognition tasks* reveal fundamental weaknesses. A close inspection reveals poor performance: most LoRA-based models default to majority-class answers, reflecting strong language priors and insufficient adaptation to subtle visual cues. Only the fully fine-tuned RadFM reliably overcomes this bias (see Appendix G for a sensitivity/specificity analysis).

c. *Visual and Medical Reasoning* remain the most challenging. Models show inconsistent spatial reasoning, succeeding at coarse tasks (*e.g.*, , finding the largest lesion diameter) but failing at fine-grained ones (*e.g.*, , kidney volume comparison). Critically, all models falter on our simplified, clinically-grounded medical reasoning questions, signaling a significant gap in their readiness for real-world diagnostics.

**Impact of Model Architecture and Training:** Our results show that VLMs' training data size and architecture design are important for their performance on CTLesionVQA.

a. *Data scale and fine-tuning* matter more than LLM size. RadFM's superior performance is attributable to its large-scale pretraining (16M 2D+3D pairs) and full-model fine-tuning. Conversely, the smaller M3D (Phi-3-4B) outperforms its larger LLaMA2-7B variant, suggesting that architectural choices can be more impactful than parameter count alone (Yousri & Safwat, 2023).

b. *Vision architecture* is critical. Merlin adopts a 3D ResNet with a single global token projected via a linear layer, resulting in inferior performance. In contrast, RadFM, M3D, and CT-CHAT use ViT-style 3D encoders that produce token sequences, enabling richer spatial reasoning through attention. Token-level granularity appears essential for capturing complex volumetric patterns.

c. *Input resolution* shows weak correlation with performance. RadFM and M3D resize raw CT volumes directly, whereas Merlin and CT-CHAT resample spacing and crop or pad to target dimensions.

Table 3: Performance for five VLMs under multi-choice and free-text settings. Subtypes marked with * indicate free-text numerical answers evaluated using MRA, higher is better. Meas. = Measurement, Recog. = Recognition, Vis. Rsn. = Visual Reasoning, Med. Rsn. = Medical Reasoning.

| Type | Subtype | Multi-choice | | | | | | Free-text | | | | |
| | | Rand | Merlin | M3D-L2 | M3D-P3 | CT-CHAT | RadFM | Merlin | M3D-L2 | M3D-P3 | CT-CHAT | RadFM |
|---|---|---|---|---|---|---|---|---|---|---|---|---|
| Meas. | lesion volume measurement* | 0.250 | 0.253 | 0.815 | 0.825 | 0.833 | 0.815 | 0.079 | 0.085 | 0.079 | 0.075 | 0.112 |
| | organ HU measurement* | 0.250 | 0.254 | 0.638 | 0.640 | 0.637 | 0.647 | 0.487 | 0.490 | 0.491 | 0.513 | 0.608 |
| | organ volume measurement* | 0.250 | 0.262 | 0.747 | 0.754 | 0.750 | 0.755 | 0.526 | 0.535 | 0.528 | 0.549 | 0.583 |
| | **Average** | 0.250 | 0.256 | 0.733 | **0.740** | **0.740** | 0.739 | 0.364 | 0.370 | 0.366 | 0.379 | **0.434** |
| Recog. | colon lesion existence | 0.500 | 0.859 | 0.859 | 0.859 | 0.859 | 0.856 | 0.859 | 0.859 | 0.859 | 0.859 | 0.893 |
| | kidney cyst existence | 0.500 | 0.797 | 0.797 | 0.797 | 0.797 | 0.861 | 0.797 | 0.797 | 0.797 | 0.797 | 0.864 |
| | kidney lesion existence | 0.500 | 0.495 | 0.510 | 0.501 | 0.514 | 0.668 | 0.511 | 0.515 | 0.490 | 0.507 | 0.692 |
| | kidney tumor existence | 0.500 | 0.564 | 0.574 | 0.574 | 0.574 | 0.886 | 0.574 | 0.574 | 0.574 | 0.574 | 0.890 |
| | liver lesion existence | 0.500 | 0.535 | 0.524 | 0.517 | 0.524 | 0.652 | 0.524 | 0.524 | 0.524 | 0.524 | 0.662 |
| | pancreatic lesion existence | 0.500 | 0.718 | 0.718 | 0.718 | 0.718 | 0.810 | 0.718 | 0.718 | 0.718 | 0.718 | 0.871 |
| | **Average** | 0.500 | 0.661 | 0.664 | 0.661 | 0.664 | **0.789** | 0.664 | 0.665 | 0.660 | 0.663 | **0.812** |
| Vis. Rsn. | adjacent organ | 0.333 | 0.217 | 0.565 | 0.609 | 0.609 | 0.609 | 0.174 | 0.174 | 0.304 | 0.304 | 0.435 |
| | inter-segment comparison | 0.333 | 0.470 | 0.567 | 0.576 | 0.572 | 0.591 | 0.577 | 0.561 | 0.592 | 0.589 | 0.456 |
| | kidney volume comparison | 0.333 | 0.347 | 0.370 | 0.364 | 0.372 | 0.386 | 0.350 | 0.370 | 0.356 | 0.370 | 0.386 |
| | largest lesion attenuation | 0.333 | 0.317 | 0.541 | 0.539 | 0.544 | 0.555 | 0.526 | 0.544 | 0.548 | 0.542 | 0.521 |
| | largest lesion diameter* | 0.250 | 0.263 | 0.778 | 0.783 | 0.781 | 0.766 | 0.182 | 0.209 | 0.233 | 0.269 | 0.232 |
| | largest lesion location | 0.392 | 0.307 | 0.310 | 0.310 | 0.340 | 0.340 | 0.359 | 0.353 | 0.337 | 0.353 | 0.334 |
| | largest lesion slice* | 0.250 | 0.241 | 0.672 | 0.684 | 0.672 | 0.664 | 0.524 | 0.533 | 0.510 | 0.513 | 0.672 |
| | lesion count by location* | 0.250 | 0.583 | 0.861 | 0.860 | 0.862 | 0.861 | 0.534 | 0.534 | 0.534 | 0.534 | 0.506 |
| | lesion counting* | 0.328 | 0.455 | 0.781 | 0.784 | 0.796 | 0.790 | 0.000 | 0.000 | 0.000 | 0.000 | 0.001 |
| | lesion outlier | 0.500 | 0.521 | 0.507 | 0.549 | 0.451 | 0.493 | 0.451 | 0.535 | 0.535 | 0.577 | 0.521 |
| | liver lesion clustering | 0.333 | 0.331 | 0.438 | 0.475 | 0.463 | 0.469 | 0.388 | 0.469 | 0.469 | 0.431 | 0.513 |
| | organ aggregation* | 0.250 | 0.257 | 0.660 | 0.667 | 0.655 | 0.661 | 0.577 | 0.569 | 0.586 | 0.574 | 0.621 |
| | organ enlargement | 0.500 | 0.736 | 0.736 | 0.736 | 0.736 | 0.746 | 0.736 | 0.736 | 0.736 | 0.736 | 0.759 |
| | tumor organ HU difference* | 0.305 | 0.296 | 0.836 | 0.839 | 0.821 | 0.821 | 0.113 | 0.122 | 0.139 | 0.197 | 0.189 |
| | **Average** | 0.335 | 0.382 | 0.616 | **0.627** | 0.620 | 0.625 | 0.392 | 0.408 | 0.420 | 0.428 | **0.439** |
| Med. Rsn. | fatty liver | 0.333 | 0.318 | 0.461 | 0.455 | 0.481 | 0.481 | 0.481 | 0.481 | 0.396 | 0.487 | 0.578 |
| | lesion type classification | 0.500 | 0.865 | 0.865 | 0.865 | 0.865 | 0.865 | 0.865 | 0.865 | 0.865 | 0.865 | 0.851 |
| | pancreatic cyst resectability | 0.500 | 0.371 | 0.657 | 0.800 | 0.800 | 0.771 | 0.800 | 0.800 | 0.800 | 0.800 | 0.771 |
| | pancreatic lesion resectability | 0.333 | 0.379 | 0.483 | 0.483 | 0.483 | 0.483 | 0.414 | 0.483 | 0.483 | 0.483 | 0.483 |
| | pancreatic steatosis | 0.500 | 0.526 | 0.526 | 0.513 | 0.513 | 0.579 | 0.526 | 0.526 | 0.526 | 0.526 | 0.658 |
| | pancreatic tumor staging | 0.250 | 0.216 | 0.351 | 0.243 | 0.189 | 0.324 | 0.216 | 0.216 | 0.297 | 0.135 | 0.432 |
| | **Average** | 0.403 | 0.446 | 0.557 | 0.560 | 0.555 | **0.584** | 0.550 | 0.562 | 0.561 | 0.549 | **0.629** |
| | **Total Average** | 0.369 | 0.440 | 0.626 | 0.632 | 0.628 | **0.662** | 0.478 | 0.489 | 0.493 | 0.497 | **0.555** |

In theory, spacing inconsistency may degrade volume-sensitive measurements, yet we observe that direct resizing does not hurt performance on tasks such as *organ volume measurement*, *lesion volume measurement*, and *kidney volume comparison*. Similarly, CT-CHAT receives the largest input size ([300, 300, 600]) but underperforms across most reasoning tasks. Merlin processes [224, 224, 160] volumes but yields even lower overall accuracy.

## 4.3 Impact of Measurement and Recognition on Reasoning Tasks.

We train RadFM without measurement and recognition tasks to see whether there is a crucial impact of basic tasks on higher-level tasks. The relatively small performance gap in Fig. 4 suggests that RadFM already generalizes reasonably well to reasoning tasks, regardless of whether measurement/recognition is explicitly seen during training. We hypothesize the main reason is that RadFM was pre-trained on large-scale 2D/3D image-text data, including structured reports, which may implicitly cover recognition and measurement concepts. But we still find that in several subtypes like *inter-segment comparison* and *tumor organ HU difference*, combined training brings a notable benefit. These subtypes heavily rely on the annotation of liver subsegments and HU values in our dataset.

## 4.4 Impact of Lesion Properties on Recognition Performance.

To better understand the factors that influence lesion recognition performance, we analyze RadFM's recognition sensitivity across different lesion sizes and HU contrasts. Figure 5 (left) shows sensitivity grouped by lesion diameter (<2cm, 2–4cm, >4cm). For kidney tumors, sensitivity increases with size. However, liver and pancreatic lesions do not follow this trend; in particular, sensitivity for large pancreatic lesions decreases. We hypothesize that this may stem from anatomical complexity

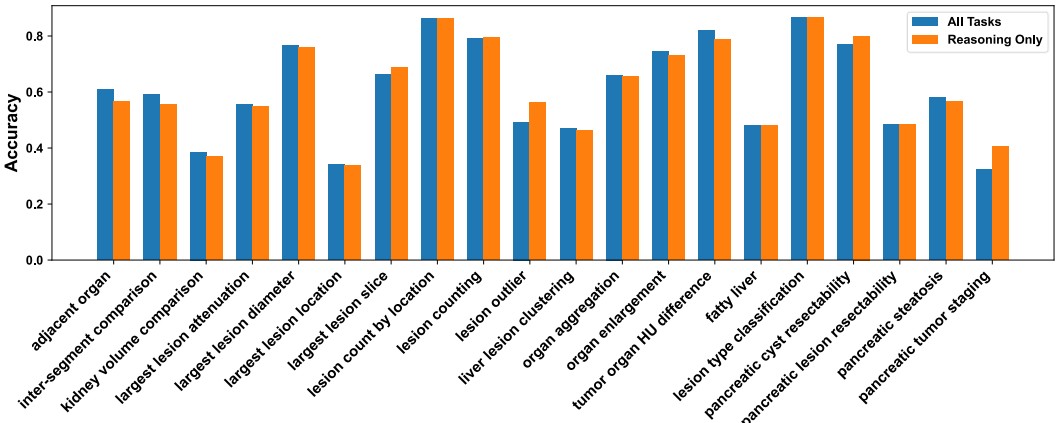

Figure 4: The RadFM accuracy of reasoning tasks with or without measurement/recognition tasks.

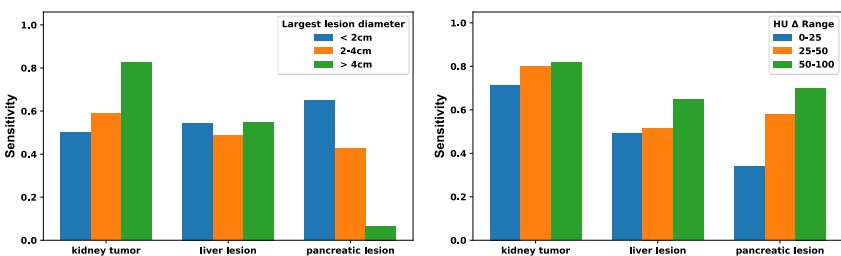

Figure 5: Lesion recognition sensitivity of RadFM under different lesion sizes (left) and HU contrast ranges (right). **Left:** Sensitivity increases with size only for kidney tumors, while liver and pancreatic lesions show no consistent trend. **Right:** Higher HU contrast leads to higher sensitivity across all lesion types, indicating that intensity-based features significantly affect detection performance.

obscuring large lesions, annotation imbalance, or model reliance on contextual rather than absolute size cues. In contrast, Figure 5 (right) shows a consistent increase in sensitivity with larger lesion-to-organ HU differences (0–25, 25–50, 50–100). This trend holds across all lesion types and suggests that stronger intensity contrast enhances boundary detectability, making HU difference a more reliable predictor of VLM sensitivity than physical size. We also analyze the impact of potential confounders such as scanner variation, sex subgroups, and other demographic factors in Appendix H.

### 4.5 Improving Lesion Recognition via Segmentation-based Preprocessing.

Despite their success on general VQA tasks, current VLMs exhibit substantial failures in lesion recognition, especially for small ones. As shown in Table 4, several models (*e.g.*, M3D-LLaMA2, M3D-Phi3, CT-CHAT) collapse into predicting the dominant class across all samples, leading to imbalanced sensitivity and specificity. This indicates that without explicit spatial localization, VLMs fail to attend to subtle lesion signals in raw 3D volumes. To address this, we propose a simple yet effective strategy that crops the input images around target organs through nnUNet (Isensee et al., 2021) anatomical localization. This approach reduces the noisy information of irrelevant regions and zooms in on target organs. We denote the resulting models as **nnVLM** variants.

Our experiments in Table 4 show that nnM3D achieves substantial gains in lesion recognition across all three organs. For instance, nnM3D-LLaMA2 improves kidney tumor sensitivity from 0% to 80.9%, surpassing even RadFM in this case. These results highlight the importance of anatomical context in vision-language learning, and suggest that simple localization priors can serve as effective alternatives to full voxel-level supervision. Figure 6 further visualizes model-level performance using Youden's index. While liver and pancreas remain challenging, multiple VLMs approach or match the oracle's performance on kidney tumors. This suggests that with targeted preprocessing, medical

Table 4: Lesion recognition sensitivity, specificity, and accuracy (%) for three organs across nnUNet (oracle), existing VLMs, and our proposed nnM3D that uses nnUNet for organ localization.

| Model | Liver Lesion | | | Kidney Tumor | | | Pancreatic Lesion | | |
|---|---|---|---|---|---|---|---|---|---|
| | Sens. | Spec. | Acc. | Sens. | Spec. | Acc. | Sens. | Spec. | Acc. |
| nnUNet (oracle) | 86.2 | 73.4 | 81.7 | 96.3 | 78.3 | 87.7 | 80.0 | 76.6 | 78.9 |
| RadFM | 53.0 | 78.6 | 65.2 | 75.2 | 98.6 | 88.6 | 40.3 | 97.0 | 81.0 |
| M3D-Phi3 | 90.8 | 8.7 | 51.7 | 0.0 | 100.0 | 57.4 | 0.0 | 100.0 | 71.8 |
| M3D-LLaMA2 | 100.0 | 0.0 | 52.4 | 0.0 | 100.0 | 57.4 | 0.0 | 100.0 | 71.8 |
| Merlin | 52.7 | 52.0 | 52.4 | 50.2 | 51.2 | 50.7 | 48.9 | 51.6 | 50.8 |
| CT-CHAT | 100.0 | 0.0 | 52.4 | 0.0 | 100.0 | 57.4 | 0.0 | 100.0 | 71.8 |
| nnM3D-Phi3 | 63.7 | 62.0 | 62.9 | 79.1 | 95.7 | 88.6 | 2.6 | 98.2 | 71.3 |
| nnM3D-LLaMA2 | 67.6 | 58.6 | 66.3 | 80.9 | 95.3 | 89.2 | 35.1 | 91.6 | 75.7 |

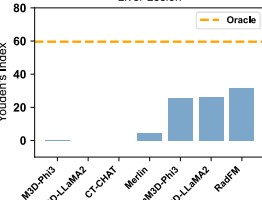
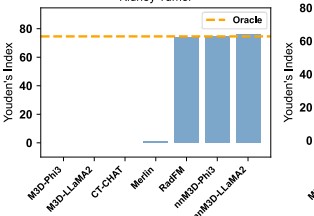
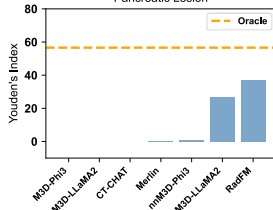

Figure 6: Comparison of Youden's Index (sensitivity + specificity - 1) of VLMs and the oracle.

VLMs may close the gap with segmentation-based recognition methods. We expect our benchmark can witness VLMs' improvement, getting closer or even surpassing the segmentation methods.

## 5 DISCUSSION AND CONCLUSION

**Are 3D medical VLMs precise and intelligent enough for clinical diagnosis?** This work takes a step toward answering this question by introducing **CTLesionVQA**, the first large-scale VQA benchmark focused on 3D clinical diagnosis that enables not only quantitative evaluation but also tracing of model failures. Through extensive evaluation, our dataset reveals that while VLMs exhibit emerging precision in basic measurement and recognition (even approaching segmentation models), their overall intelligence remains far from meeting clinical requirements, especially in medical reasoning tasks. Through careful inspection, we reveal the impact of basic tasks on the reasoning task, and also analyze the difficulty of lesion recognition *w.r.t.* both lesion size and HU contrast.

**The dataset exposes critical differences in 3D medical VLMs.** First, visual architectures matter: our experiments show that ViT-based 3D encoders significantly outperform single-token CNN backbones in tasks requiring spatial reasoning or multi-lesion aggregation. Second, language decoder scale alone does not guarantee improved performance; rather, large-scale pretraining and full-tuning strategies (*e.g.*, RadFM) yield more consistent gains across tasks. Third, our proposed organ-specific preprocessing pipeline demonstrates that vision models with anatomical priors significantly improve lesion detection by spatial localization.

**Limitations.** The dataset construction process relies heavily on precise organ and lesion segmentation to generate structured metadata and QA pairs. However, due to the inherent variability in radiologist expertise and the ambiguity of certain CT appearances (*e.g.*, low-contrast lesions or anatomical variants), the imperfect segmentation quality may introduce noise into downstream QA pairs. The dataset is intended as a research benchmark, not for clinical deployment or decision-making that may cause risks for false reassurance or missed diagnoses. Additionally, while our experiments provide insightful comparisons across VLM architectures and training regimes, the conclusions would benefit from more controlled ablation studies to isolate variables systematically.

**Conclusion.** CTLesionVQA fills a critical gap in the evaluation of medical VLMs. It serves both as a diagnostic tool and as a development benchmark. We will hold recurring challenges to support the community in building safer, more explainable, and ultimately clinically useful multimodal systems.

ETHICS STATEMENT

The development and evaluation of the CTLesionVQA benchmark were conducted with careful consideration of ethical principles, particularly concerning medical data and the potential impact of diagnostic AI systems.

**Data and Privacy:** The CTLesionVQA dataset is constructed exclusively from 17 publicly available, de-identified medical imaging datasets. We did not collect any new patient data for this work. Our data curation process included a thorough verification step to ensure that, to the best of our knowledge, no personally identifiable information (PII) is present in the images or associated metadata we release. The intended use of this dataset is strictly for research purposes within the academic and industrial research communities to advance medical AI.

**Intended Use and Potential Misuse:** The primary purpose of this work is to provide a rigorous benchmark to evaluate the current capabilities and limitations of Vision-Language Models for clinical diagnosis. **It is not intended for direct clinical use, nor should any models trained on this benchmark be deployed in a clinical setting without extensive further validation and regulatory approval.** A key finding of our paper is that current VLMs are *not* yet ready for complex clinical diagnostics. We believe that by transparently highlighting these limitations, our work helps mitigate the risk of premature deployment of unreliable AI systems. A potential misuse could involve overstating the capabilities of models based on their performance on this benchmark, leading to false confidence in their diagnostic accuracy.

**Bias and Fairness:** The dataset is aggregated from 88 different centers, which provides a degree of diversity in terms of scanners and acquisition protocols. However, the underlying demographic distributions (*e.g.*, age, sex, ethnicity) are inherited from the source datasets and may not be fully representative of the global population. While we provide a breakdown of performance across age and sex in Appendix H, we acknowledge that models trained on CTLesionVQA may exhibit biases. We encourage future research to use our benchmark to study and mitigate these biases.

**Annotator Labor:** All 23 radiologists who participated in the annotation process were fairly compensated for their expert labor. The annotation protocol was designed by senior specialists to ensure high quality and consistency, as detailed in Appendix C.

REPRODUCIBILITY STATEMENT

We are committed to ensuring the full reproducibility of our research. All data, code, and experimental details will be made publicly available.

**Data Availability:** The CTLesionVQA benchmark, including all 395K question-answer pairs and the corresponding metadata, will be released under a permissive license. We will provide download links and detailed instructions in a public GitHub repository. The list of the 17 public source datasets used to construct CTLesionVQA is provided in Table 1, and our repository will include scripts and links to access this source data.

**Code Availability:** We will release all code necessary to reproduce our experiments in the same GitHub repository. This includes:

1. The complete data processing and question-answer generation pipeline.

2. Evaluation scripts to compute all reported metrics (Accuracy, MRA, Sensitivity, Specificity).

3. Training and inference code for all four benchmarked models (RadFM, M3D, Merlin, and CT-CHAT), adapted for the CTLesionVQA dataset.

**Experimental Details:** All hyperparameters, model architectures, and training configurations are detailed in Appendix E (Table 9). This includes learning rates, batch sizes, optimizer details, and the specific open-source codebases we adapted. The computational resources used (NVIDIA A5000, A6000, and A100 GPUs) and approximate training times (48 hours per model) are also specified to allow for accurate replication of the experimental setup. We will also release the fine-tuned model weights for all benchmarked models to facilitate further research.

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

# A  METADATA AND STRUCTURED DESCRIPTION GENERATION

To support systematic question generation and fine-grained model evaluation, we construct a rich set of structured metadata for each CT volume using paired organ and lesion segmentation masks. This section describes how we derive the metadata fields and generate a radiology-style structured description for each case.

## A.1  METADATA EXTRACTION FROM SEGMENTATION MASKS

Given a CT volume and corresponding 3D segmentation masks for organs and lesions, we extract anatomical and lesion-level statistics through the following steps:

- **Resampling and alignment:** All masks are resampled to the same voxel spacing as the CT image. We ensure alignment across volumes and segmentations to preserve geometric correctness.

- **Volume and size statistics:** For each organ and its lesions (*e.g.*, liver, kidney, pancreas), we compute total organ volume, total lesion volume, and the number of lesion instances.

- **Largest lesion analysis:** We extract the size (diameter), location (*e.g.*, liver segment or organ side), and mean attenuation (HU value) of the largest lesion per organ and subtype (tumor, cyst, or unspecified lesion).

- **Enhancement type classification:** Using the HU value difference between lesions and organ parenchyma, we classify lesion attenuation into three categories: *hyperattenuating*, *isoattenuating*, and *hypoattenuating*.

- **Clinical staging:** For pancreas tumors, we approximate T-stage (T1–T4) based on existing staging protocols.

- **Demographic and acquisition metadata:** Patient age, sex, contrast phase, and scanner type are retrieved from DICOM headers or accompanying metadata files.

The final metadata table includes over 70 structured attributes per scan, such as: *liver lesion count, largest kidney tumor diameter (cm), pancreatic tumor attenuation, spleen volume, organ HU values, lesion location,* and more. This table enables compositional and interpretable question generation across a wide range of diagnostic concepts.

## A.2  STRUCTURED REPORT-STYLE DESCRIPTION

In addition to the tabular metadata, we generate a structured textual description in radiology report style for each scan following (Bassi et al., 2025). This free-text summary provides high-resolution lesion-level information and mimics real radiological narratives. Each description includes:

- A global summary per organ (*e.g.*, volume, mean HU).

- Instance-level lesion summaries: lesion size, volume, location (*e.g.*, liver segment, pancreas head/body/tail), slice number, attenuation classification.

- Aggregated lesion counts and total tumor/cyst volumes.

- Impression statement summarizing major findings, such as: *"Multiple (25) hypoattenuating liver masses. Largest one (segment 2) measures 3.2 x 1.7 cm. Total volume of all liver masses: 19.4 cm³."*

An example of the structured description is shown in the following.

---
**Example: Radiology-style Structured Report**

CT Arterial Phase

FINDINGS:

---

Liver: Normal size (volume: 1293.7 cm³). Mean HU value: 111.3 ± 17.4.

Liver lesions: Liver lesion 1: Location: hepatic segment 2. Size: 3.2 x 1.7 cm (image 174). Volume: 8.1 cm³. Enhancement relative to liver: Hypoattenuating (HU value is 9.6 ± 19.8).

Liver lesion 2: Location: hepatic segment 8. Size: 2.5 x 2.0 cm (image 178). Volume: 4.9 cm³. Enhancement relative to liver: Hypoattenuating (HU value is 36.3 ± 31.3).

... [truncated for brevity]

Liver lesion 24: Location: hepatic segment 5. Size: 0.5 x 0.4 cm (image 156). Volume: 0.1 cm³. Enhancement relative to liver: Hypoattenuating (HU value is 97.2 ± 16.9).

Liver lesion 25: Location: hepatic segment 4. Size: 0.4 x 0.3 cm (image 156). Volume: 0.0 cm³. Enhancement relative to liver: Hypoattenuating (HU value is 71.3 ± 20.0).

Pancreas: Normal size (volume: 80.3 cm³). Mean HU value: 104.8 ± 28.5.

Kidney: Normal size (right kidney volume: 166.6 cm³; left kidney volume: 156.6 cm³; total kidney volume: 323.2 cm³). Mean HU value: 127.4 ± 52.8.

Spleen: Normal size (volume: 135.1 cm³). Mean HU value: 124.7 ± 34.8.

IMPRESSION: Multiple (25) hypoattenuating liver masses. Largest one (hepatic segment 2) measures 3.2 x 1.7 cm. Total volume of all liver masses: 19.4 cm³.

These descriptions support reasoning question generation (*e.g.*, "How are the liver lesions distributed across hepatic segments") and provide explainable context for model output interpretation.

## B  TASK DEFINITIONS AND GENERATION LOGIC

To support a systematic and diverse evaluation of VLMs in 3D lesion-centric diagnosis, CTLesion-VQA includes 29 question subtypes spanning four diagnostic categories: *Measurement*, *Recognition*, *Visual Reasoning*, and *Medical Reasoning*.

Each question subtype corresponds to a well-defined clinical concept (e.g., organ size, lesion count, resectability), and is generated through a rule-based or metadata-driven functional program. These question types are designed to reflect increasing levels of diagnostic complexity, ranging from direct retrieval to multi-step inference.

Table 5 summarizes all subtypes, their task type, the logic used for answer generation, and an example QA pair. This structured taxonomy enables reproducible benchmarking and compositional analysis of VLM performance across clinical tasks.

Table 5: Summary of task types, subtypes, generation logic, and example question-answer pairs in **CTLesionVQA**. Tasks are organized by their diagnostic intent: measurement, recognition, visual reasoning, and medical reasoning.

| Task Type | Subtype | Definition / Generation Logic | Example QA Pair |
|---|---|---|---|
| Measurement | organ volume measurement | Quantify organ size from metadata using volume. | Q: What is the liver volume? A: 1293.7 cm³ |
| Measurement | organ HU measurement | Extract organ mean HU value using regex from report. | Q: What is the mean HU of the pancreas? A: 104.8 |
| Measurement | lesion volume measurement | Sum total lesion volume from metadata (per lesion type and organ). | Q: What is the total tumor volume in the right kidney? A: 17.5 cm³ |

*(continued from previous page)*

| Task Type | Subtype | Definition / Generation Logic | Example QA Pair |
|---|---|---|---|
| Recognition | liver lesion existence | Check presence of any lesion in liver by total volume > 0. | Q: Is there any lesion in the liver? A: Yes |
| Recognition | pancreatic lesion existence | Check presence of any lesion in pancreas. | Q: Does the pancreas have any lesions? A: No |
| Recognition | kidney lesion existence | Check presence of non-specific kidney lesions. | Do we have evidence of any lesions within the kidney? A: No |
| Recognition | kidney cyst existence | Check presence of cyst in kidney. | Is there at least one cyst in the kidney? A: No |
| Recognition | kidney tumor existence | Check presence of tumor in kidney. | Would the kidney be described as having tumors? A: Yes |
| Recognition | colon lesion existence | Check presence of colon lesions. | Is the colon affected by any lesions? A: No |
| Visual Reasoning | lesion counting | Count lesion instances by type and organ. | Q: How many cysts are there in the liver? A: 3 |
| Visual Reasoning | largest lesion diameter | Use metadata field for largest lesion diameter. | Q: What is the diameter of the largest tumor in the pancreas? A: 2.5 cm |
| Visual Reasoning | largest lesion location | Read lesion location label (e.g. segment 1–8 or left/right). | Q: Where is the largest liver lesion located? A: Segment 2 |
| Visual Reasoning | largest lesion attenuation | Classify lesion HU vs. background as hypo/iso/hyper. | Q: Is the largest liver cyst hypoattenuating? A: Yes |
| Visual Reasoning | kidney volume comparison | Compare left/right kidney volumes and discretize into 3 options. | Q: Which kidney is larger? A: Left kidney |
| Visual Reasoning | organ aggregation | Sum two organs' volumes. | Q: What is the combined volume of liver and spleen? A: 1428.3 cm³ |
| Visual Reasoning | tumor organ HU difference | Compute absolute HU diff between lesion and corresponding organ. | Q: What is the HU difference between kidney tumor and kidney? A: 32.4 |
| Visual Reasoning | largest lesion slice | Locate axial slice with max lesion size and normalize by depth. | Q: On which slice is the largest liver lesion found? A: Slice 174 |
| Visual Reasoning | lesion outlier | If largest lesion is >3× volume of second largest → outlier. | Q: Is the largest lesion 3× larger than the second largest? A: No |
| Visual Reasoning | lesion count by location | Extract per-segment or sub-region lesion counts from report. | Q: How many liver lesions are in segment 8? A: 5 |
| Visual Reasoning | inter-segment comparison | Compare lesion counts between two liver segments. | Q: Which segment has more lesions: segment 2 or 4? A: Segment 2 |
| Visual Reasoning | adjacent organ | Extract from text: reported adjacent organ names for largest lesion. | Q: Which organ is adjacent to the largest liver lesion? A: Stomach |
| Visual Reasoning | organ enlargement | Use 'enlarged' keyword from report per organ. | Q: Is the pancreas enlarged? A: No |
| Visual Reasoning | liver lesion clustering | If > 3 lesions within 3 adjacent segments, mark as 'clustered'. | Q: Are liver lesions clustered in adjacent segments? A: Yes |
| Medical Reasoning | pancreatic tumor staging | Use labeled T-stage for pancreatic tumor. | Q: What is the stage of the pancreatic tumor? A: T2 |
| Medical Reasoning | fatty liver | Use liver/spleen HU ratio and liver HU to classify steatosis severity. | Q: Does the liver show fatty infiltration? A: Yes |
| Medical Reasoning | pancreatic steatosis | Use pancreas/spleen HU ratio to assess steatosis (<0.7 = Yes). | Q: Does the pancreas show steatosis? A: No |
| Medical Reasoning | pancreatic cyst resectability | Binary classification: cyst volume > 3.0 cm³ → resection. | Q: Is the pancreatic cyst resectable? A: Yes |
| Medical Reasoning | lesion type classification | If largest kidney lesion HU > threshold → tumor else cyst. | Q: Is the kidney lesion a cyst or tumor? A: Tumor |
| Medical Reasoning | pancreatic lesion resectability | Use largest lesion's report-tagged resectability field. | Q: Can the pancreatic lesion be surgically resected? A: No |

## C    DETAILS FOR THE ANNOTATORS

A full table for the radiologists is provided in Table 6 for transparency. Annotator backgrounds range from senior board-certified abdominal imaging specialists (*e.g.*, S3: 35 years experience) to general radiologists and residents with intensive CT reading experience (up to 18,000 CTs/year).

To ensure annotation reliability:

- Initial segmentation were primarily performed by general radiologists and residents.

- Board-certified specialists oversaw the process by double-checking all annotations, adjudicating ambiguous cases, and ensuring anatomical and diagnostic consistency. They served as clinical leads, akin to project managers for annotation quality control.

Table 6: Detailed Information of the 23 Annotators.

| No. | Annotator ID | Experience (yr) | CT read / year |
|-----|-------------|----------------|----------------|
| 1 | Specialist 1 (S1) | 24 | 12,000 |
| 2 | Specialist 2 (S2) | 22 | 12,000 |
| 3 | Specialist 3 (S3) | 35 | 8,000 |
| 4 | Specialist 4 (S4) | 30 | 8,000 |
| 5 | Specialist 5 (S5) | 28 | 9,000 |
| 6 | Specialist 6 (S6) | 19 | 13,000 |
| 7 | Specialist 7 (S7) | 23 | 11,000 |
| 8 | General 1 (G1) | 12 | 18,000 |
| 9 | General 2 (G2) | 8 | 18,000 |
| 10 | General 3 (G3) | 9 | 18,000 |
| 11 | General 4 (G4) | 10 | 18,000 |
| 12 | General 5 (G5) | 8 | 18,000 |
| 13 | General 6 (G6) | 13 | 18,000 |
| 14 | General 7 (G7) | 11 | 18,000 |
| 15 | General 8 (G8) | 10 | 18,000 |
| 16 | General 9 (G9) | 10 | 18,000 |
| 17 | General 10 (G10) | 13 | 18,000 |
| 18 | General 11 (G11) | 10 | 18,000 |
| 19 | Resident 1 (R1) | 5 | 16,000 |
| 20 | Resident 2 (R2) | 3 | 16,000 |
| 21 | Resident 3 (R3) | 4 | 16,000 |
| 22 | Resident 4 (R4) | 5 | 16,000 |
| 23 | Resident 5 (R5) | 5 | 16,000 |

## D    EVALUATION OF SEGMENTATION QUALITY

To validate the quality of the annotations used for generating the CTLesionVQA dataset, we conducted a rigorous evaluation by training both organ and lesion segmentation models on our data. The following sections detail the out-of-distribution (OOD) performance on external, unseen datasets.

### D.1    ORGAN SEGMENTATION QUALITY

We trained a U-Net model on the organ masks from CTLesionVQA and evaluated its generalization performance on the testing sets of two external datasets: TotalSegmentator and an internal Johns Hopkins Hospital (JHH) dataset. Table 7 compares the Dice Similarity Coefficient (DSC) of the model trained on our annotations ("ours") against a model trained on the in-distribution ("iid") data of the respective benchmark. The results demonstrate that our annotations produce models with strong generalization capabilities.

Table 7: Organ Segmentation Performance (DSC, %) on External Datasets.

| Organ | TotalSegmentator (ours) | TotalSegmentator (iid) | JHH (ours) | JHH (iid) |
|---|---|---|---|---|
| Spleen | 95.2 ± 0.0 | 98.4 | 95.0 ± 0.0 | 97.1 |
| Kidney (R) | 92.5 ± 0.2 | 94.7 | 92.2 ± 0.0 | 98.4 |
| Kidney (L) | 89.0 ± 0.3 | 94.4 | 91.6 ± 0.1 | 96.1 |
| Gall bladder | 82.8 ± 0.2 | 84.5 | 83.6 ± 0.2 | 90.5 |
| Liver | 94.7 ± 0.2 | 96.3 | 95.0 ± 0.3 | 98.6 |
| Stomach | 85.2 ± 0.3 | 95.5 | 85.0 ± 0.2 | 96.7 |
| Aorta | 75.6 ± 0.2 | 98.2 | 73.9 ± 0.3 | 98.2 |
| IVC | 74.2 ± 0.2 | 93.4 | 77.7 ± 0.4 | 90.8 |
| Pancreas | 83.5 ± 0.2 | 89.4 | 80.9 ± 0.6 | 87.8 |

### D.2 LESION SEGMENTATION AND DETECTION QUALITY

To assess the quality of our lesion annotations, we trained a nnUNet model and evaluated its lesion detection performance on an external dataset. Table 8 summarizes the sensitivity and specificity, stratified by lesion size. The high sensitivity, even for small lesions ($\leq$2cm), confirms that our annotations are reliable and can support models that detect clinically challenging lesions.

Table 8: Lesion Detection Performance on an External Dataset.

| Lesion Type | Sensitivity (%) | Specificity (%) |
|---|---|---|
| **Large Tumors > 2cm** | | |
| Liver Tumor (HCC) | 89.4 (269/301) | 73.4 (179/244) |
| Kidney Tumor (RCC) | 97.3 (213/219) | 73.4 (179/244) |
| Pancreas Tumor (PDAC) | 91.4 (96/105) | 76.6 (187/244) |
| **Small Tumors $\leq$2cm** | | |
| Liver Tumor (HCC) | 79.6 (113/142) | 73.4 (179/244) |
| Kidney Tumor (RCC) | 92.0 (46/50) | 78.3 (191/244) |
| Pancreas Tumor (PDAC) | 76.9 (296/385) | 76.6 (187/244) |

## E TRAINING DETAILS FOR VLMS

We provide detailed training configurations for the four benchmarked vision-language models (VLMs) evaluated in this work: RadFM, M3D (with both LLaMA2 and Phi-3 decoders), Merlin, and CT-CHAT. To ensure a fair comparison, all models are trained using their official open-source codebases and adapted to the CTLesionVQA dataset with minimal changes to architecture or optimization logic.

Table 9 summarizes key hyperparameters and compute resource settings. All models are trained with AdamW optimizer and cosine learning rate scheduling. Mixed-precision training is enabled using either FP16 or BF16, depending on framework compatibility. For large models such as RadFM and M3D, gradient accumulation is used to simulate larger batch sizes, with 4 GPUs and 16 CPU workers for data loading.

Notably, due to high memory requirements, Merlin is trained with a batch size of 1 and gradient accumulation of 8, while CT-CHAT benefits from a higher per-device batch size due to its lighter vision backbone. Training for all models is conducted for approximately 48 hours using commodity GPU clusters (NVIDIA A5000, A6000, and A100 as indicated).

## F STATISTICAL ROBUSTNESS OF BENCHMARKING RESULTS

To address the statistical robustness of our findings, we report the 95% confidence intervals for accuracy on the six binary lesion recognition tasks through 5 repeated experiments. As shown in

Table 9: Model training hyperparameters and compute resource for four benchmarked VLMs.

| Item | RadFM | M3D (LLaMA2 / Phi-3) | Merlin | CT-CHAT |
|---|---|---|---|---|
| Learning rate | 5e-6 | 5e-5 | 1e-4 | 2e-5 |
| Optimizer | AdamW (8-bit) | AdamW | AdamW | AdamW |
| Auto mixed precision | FP16 | BF16 | BF16 | FP16 |
| Per device batch size | 4 (model parallel) | 1 | 1 | 32 |
| Gradient accumulation steps | 8 | 8 | 8 | 1 |
| Learning rate scheduler | Cosine | Cosine | Cosine | Cosine |
| Warmup ratio | 0 | 0.03 | 0.03 | 0.03 |
| Training iterations | 25k | 33k | 25k | 3 epochs |
| CPU workers | 16 | 16 | 16 | 128 |
| GPU hardware | 4×A5000 24GB | 4×A6000 48GB | 4×A5000 24GB | 4×A100 80GB |
| RAM | 128GB | 128GB | 256GB | 1024GB |
| Compute time | 48 hours | 48 hours | 48 hours | 48 hours |

Tables 10 and 11, the narrow intervals across all models indicate that the performance trends reported in the main paper are stable and reliable.

Table 10: Recognition task performance with 95% confidence intervals (multi-choice).

| Question Subtype | Merlin | M3D-L2 | M3D-P3 | CT-CHAT | RadFM |
|---|---|---|---|---|---|
| colon lesion existence | 0.859 [0.852, 0.866] | 0.859 [0.852, 0.866] | 0.859 [0.852, 0.866] | 0.859 [0.852, 0.866] | 0.856 [0.849, 0.863] |
| kidney cyst existence | 0.797 [0.790, 0.804] | 0.797 [0.790, 0.804] | 0.797 [0.790, 0.804] | 0.797 [0.790, 0.804] | 0.868 [0.861, 0.875] |
| kidney lesion existence | 0.495 [0.488, 0.502] | 0.510 [0.503, 0.517] | 0.501 [0.494, 0.508] | 0.514 [0.507, 0.521] | 0.668 [0.661, 0.675] |
| kidney tumor existence | 0.564 [0.556, 0.572] | 0.574 [0.566, 0.582] | 0.574 [0.566, 0.582] | 0.574 [0.566, 0.582] | 0.683 [0.674, 0.692] |
| liver lesion existence | 0.535 [0.528, 0.542] | 0.524 [0.517, 0.531] | 0.524 [0.517, 0.531] | 0.520 [0.513, 0.527] | 0.652 [0.645, 0.659] |
| pancreatic lesion existence | 0.718 [0.710, 0.726] | 0.718 [0.710, 0.726] | 0.718 [0.710, 0.726] | 0.810 [0.802, 0.818] | 0.810 [0.802, 0.818] |

Table 11: Recognition task performance with 95% confidence intervals (free-text).

| Question Subtype | Merlin | M3D-L2 | M3D-P3 | CT-CHAT | RadFM |
|---|---|---|---|---|---|
| colon lesion existence | 0.859 [0.852, 0.866] | 0.859 [0.852, 0.866] | 0.859 [0.852, 0.866] | 0.859 [0.852, 0.866] | 0.893 [0.887, 0.899] |
| kidney cyst existence | 0.797 [0.790, 0.804] | 0.797 [0.790, 0.804] | 0.797 [0.790, 0.804] | 0.797 [0.790, 0.804] | 0.871 [0.865, 0.877] |
| kidney lesion existence | 0.511 [0.504, 0.518] | 0.515 [0.508, 0.522] | 0.490 [0.483, 0.497] | 0.507 [0.500, 0.514] | 0.692 [0.685, 0.699] |
| kidney tumor existence | 0.574 [0.566, 0.582] | 0.574 [0.566, 0.582] | 0.574 [0.566, 0.582] | 0.574 [0.566, 0.582] | 0.890 [0.885, 0.895] |
| liver lesion existence | 0.524 [0.517, 0.531] | 0.524 [0.517, 0.531] | 0.524 [0.517, 0.531] | 0.524 [0.517, 0.531] | 0.662 [0.655, 0.669] |
| pancreatic lesion existence | 0.718 [0.710, 0.726] | 0.718 [0.710, 0.726] | 0.718 [0.710, 0.726] | 0.718 [0.710, 0.726] | 0.871 [0.865, 0.877] |

## G   SENSITIVITY&SPECIFICITY BREAKDOWN OF BINARY RECOGNITION

For binary recognition tasks (such as "liver lesion existence"), where one class (*e.g.*, "No lesion") is frequently the majority, accuracy can be a misleading indicator. So we provide a more comprehensive breakdown of sensitivity and specificity for multi-choice (Table 12) and free-text (Table 13) questions.

Table 12: Sensitivity and Specificity for Multi-choice binary recognition subtypes. This table provides a detailed breakdown of model performance on binary classification tasks, addressing the potential for accuracy to be misleading in cases of class imbalance.

| Question Subtype | Metric | Merlin | M3D-Llama2 | M3D-Phi3 | CT-CHAT | RadFM |
|---|---|---|---|---|---|---|
| liver lesion existence | Sensitivity | 0.825 | 1.000 | 0.908 | 1.000 | 0.530 |
| | Specificity | 0.215 | 0.000 | 0.087 | 0.000 | 0.786 |
| pancreatic lesion existence | Sensitivity | 0.000 | 0.000 | 0.000 | 0.000 | 0.403 |
| | Specificity | 1.000 | 1.000 | 1.000 | 1.000 | 0.970 |
| colon lesion existence | Sensitivity | 0.000 | 0.000 | 0.000 | 0.000 | 0.100 |
| | Specificity | 1.000 | 1.000 | 1.000 | 1.000 | 0.980 |
| kidney lesion existence | Sensitivity | 0.529 | 0.900 | 0.515 | 0.900 | 0.502 |
| | Specificity | 0.459 | 0.097 | 0.487 | 0.104 | 0.844 |
| kidney tumor existence | Sensitivity | 0.075 | 0.000 | 0.000 | 0.000 | 0.752 |
| | Specificity | 0.927 | 1.000 | 1.000 | 1.000 | 0.986 |
| kidney cyst existence | Sensitivity | 0.000 | 0.000 | 0.000 | 0.000 | 0.478 |
| | Specificity | 1.000 | 1.000 | 1.000 | 1.000 | 0.958 |

Table 13: Sensitivity and Specificity for free-text binary recognition subtypes. This complements the multi-choice results, showing model performance on open-ended binary questions where class imbalance can also affect interpretation of accuracy.

| Question Subtype | Metric | Merlin | M3D-Llama2 | M3D-Phi3 | CT-CHAT | RadFM |
|---|---|---|---|---|---|---|
| liver lesion existence | Sensitivity | 1.000 | 1.000 | 1.000 | 1.000 | 0.673 |
| | Specificity | 0.000 | 0.000 | 0.000 | 0.000 | 0.651 |
| pancreatic lesion existence | Sensitivity | 0.000 | 0.000 | 0.000 | 0.000 | 0.689 |
| | Specificity | 1.000 | 1.000 | 1.000 | 1.000 | 0.943 |
| kidney lesion existence | Sensitivity | 0.900 | 1.000 | 0.100 | 0.900 | 0.661 |
| | Specificity | 0.098 | 0.000 | 0.905 | 0.091 | 0.726 |
| colon lesion existence | Sensitivity | 0.000 | 0.000 | 0.000 | 0.000 | 0.440 |
| | Specificity | 1.000 | 1.000 | 1.000 | 1.000 | 0.967 |
| kidney cyst existence | Sensitivity | 0.000 | 0.000 | 0.000 | 0.000 | 0.526 |
| | Specificity | 1.000 | 1.000 | 1.000 | 1.000 | 0.950 |
| kidney tumor existence | Sensitivity | 0.000 | 0.000 | 0.000 | 0.000 | 0.786 |
| | Specificity | 1.000 | 1.000 | 1.000 | 1.000 | 0.968 |

## H    ACCURACY BREAKDOWN ACROSS DEMOGRAPHIC AND IMAGING FACTORS

To explore whether VLM performance varies across patient or scan-related subgroups, we stratify question-answering accuracy by four categorical factors extracted from metadata: sex, age group, CT scanner manufacturer, and contrast phase. Accuracy is reported per question category: *measurement*, *recognition*, *visual reasoning*, and *medical reasoning*.

**Age.**    Figure 7 (upper left) shows that recognition and measurement tasks remain stable across most age groups, while medical reasoning accuracy is more volatile. Notably, large drops are observed in 60–69 and 90–99 bins for medical reasoning, which may reflect either smaller sample size or increased scan complexity (*e.g.*, , multi-lesion, ambiguous enhancement). This underscores the importance of stratified evaluation in medical datasets.

**Sex.**    As shown in Figure 7 (upper right), overall performance is similar across female (F) and male (M) cohorts, with no substantial gap in any task type. Recognition is the strongest category in both groups. A slight improvement in medical reasoning is observed in males, possibly due to distributional biases in training samples (*e.g.*, , sex imbalance in pancreas/uterus-related cases).

**Contrast Phase.**    As shown in Figure 7 (lower left), recognition accuracy is high and stable across all contrast phases (arterial, delay, plain, venous), suggesting robustness of perception to intensity changes. However, medical reasoning suffers in the arterial and venous phases, likely due to poor organ-lesion contrast or increased noise in attenuation-based reasoning (*e.g.*, , fatty liver, lesion enhancement).

**Scanner Manufacturer.**    In Figure 7 (lower right), all three vendors (GE, Philips, Siemens) show consistent performance on measurement and recognition tasks. However, a sharp drop in medical reasoning accuracy is observed for Siemens, potentially due to domain shift in intensity values or HU calibration differences, which may affect reasoning modules trained on scanner-agnostic data.

These results suggest that while modern VLMs can generalize well across standard factors like sex and age, their medical reasoning performance may be more sensitive to acquisition protocol and scanner variation. Future work should incorporate domain adaptation or uncertainty modeling to ensure reliability across subpopulations.

## I    THE USE OF LARGE LANGUAGE MODELS (LLMS)

During the preparation of this manuscript, a large language model (LLM) was used as a writing and editing assistant. The use of the LLM was limited to the following tasks:

- Improving grammar, clarity, and phrasing of sentences and paragraphs.

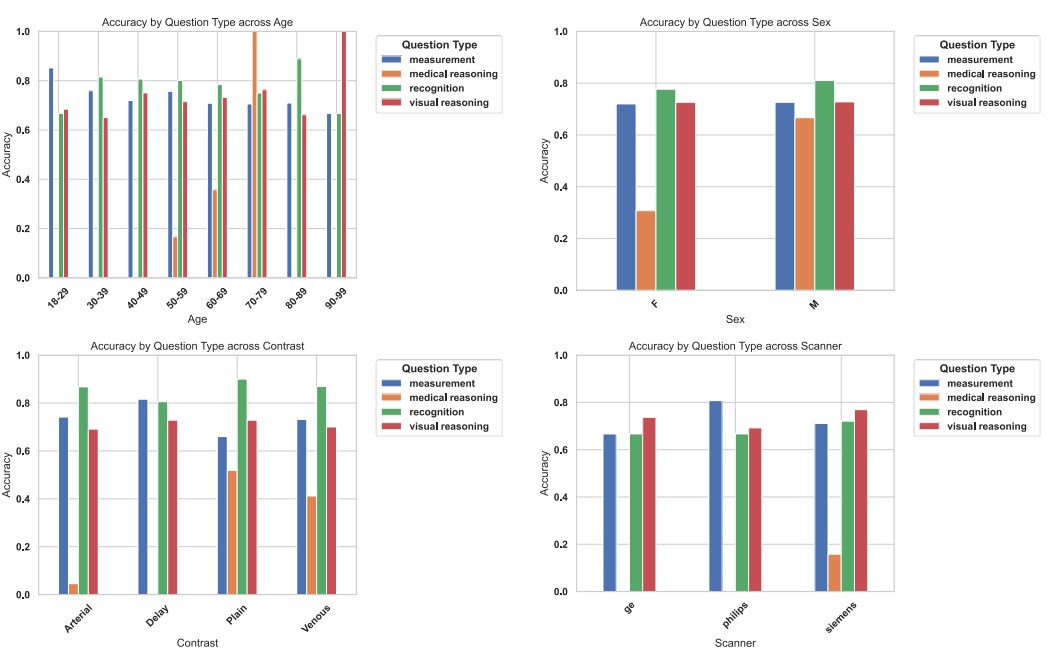

Figure 7: Accuracy by question type across Age, Sex, Contrast type, and Scanner.

- Assisting in the formatting of tables into LaTeX code.
- Compressing sections to meet length requirements.
- Reviewing the manuscript for typographical errors and inconsistencies.
- Generating boilerplate text for standard sections, such as this one.

All claims, experimental results, and scientific conclusions presented in this paper were generated by the authors. The LLM was used solely as a tool to improve the quality of the manuscript's presentation. The authors have reviewed, edited, and take full responsibility for all content in this paper.

