# OpenReview forum: "Are Vision Language Models Ready for Clinical Diagnosis? A 3D CT Benchmark for Lesion-centric Visual Question Answering"
_ICLR.cc/2026/Conference — Submitted to ICLR 2026_

### Official Review · Reviewer_NKXk · 2025-10-15

**Soundness:** 2
**Presentation:** 2
**Contribution:** 2
**Rating:** 2
**Confidence:** 4

**Summary:**

The proposes CTLesionVQA, a large-scale, lesion-centric VQA benchmark built from abdominal CT scans. The dataset aggregates 9,262 CT volumes and provides expert-designed Q&A pairs across four clinical categories: Recognition, Measurement, Visual Reasoning, and Medical Reasoning. The authors benchmark four representative 3D VLMs—RadFM, M3D, Merlin, CT-CHAT—under multiple-choice and free-text settings. Overall, models do best on measurement, but struggle on lesion recognition and especially on visual/medical reasoning, indicating they are not yet clinically ready.

**Strengths:**

The paper is well structured and written. The benchmark incorporates a large-scale CT volume dataset with lesion-centric questions.
The paper conducts a detailed analysis of model performance by contrast phase, scanner manufacturer, age, and sex to examine domain robustness. The paper does make a few observations that could be potentially interesting. For example ViT-based 3D encoders outperform CNNs on spatial reasoning. These findings are useful for guiding architecture design in future 3D VLMs.

**Weaknesses:**

1. My main concern for this paper is limited novelty. It seems more as a dataset release and benchmark than a research contribution. The question-generation approach directly repurposes CLEVR-style templates without introducing algorithmic or theoretical innovations. As a result, the work represents an incremental adaptation of existing methodologies to the medical CT domain.

2. Unfair comparisons by using models pretrained on heterogenous datasets. The authors evaluate multiple 3D VLMs on CTLesionVQA but the paper does not standardize or control for pretraining domain differences. For example, CT-CHAT is pretrained on CT-RATE, which includes non-contrast chest CT scans only. Merlin is pretrained on abdominal CT, which includes contrast-enhanced studies from a very different organs. The paper should ablate or control for pretraining modality (compare models trained on the same type of CT (e.g., abdominal contrast CT). Also the paper should explicitly analyze how pretraining dataset could affect VQA performance.

3. Medical reasoning tasks are oversimplified, by relying only on quantitative image-derived features (like HU values or lesion volumes). This oversimplifies how radiologists make diagnostic and staging decisions, which often depend on qualitative, phase-specific imaging patterns and clinical context. For Kidney lesion diagnosis, it requires multi-phase imaging and morphology (e.g., septations, wall nodules), not just HU thresholds. Pancreatic tumor staging depends on tumor–vessel contact, metastases, none of which are directly modeled in the benchmark's structured metadata. Longitudinal information is also missing to track the tumor changes overtime. Without real patient report with all these information, how can the VLM correctly predict the answer?

4. The paper lacks any description of quality checks or validation for the generated question–answer pairs. Given the reliance on rule-based templates, this omission raises serious concerns about the semantic and clinical accuracy of the dataset. Without manual auditing or domain expert review, the integrity of the benchmark’s supervision signal is uncertain, especially in healthcare setting.

5. The paper only validates organ and tumor masks, but does not verify fine-grained lesion categories (e.g., tumor vs. cyst vs. lesion). This omission is serious, as many benchmark tasks depend on accurate lesion subtyping. Without validating these distinctions, the dataset’s clinical utility for evaluating diagnostic models are fundamentally compromised.

6. The benchmark fails to include compositional or multi-step reasoning tasks, such as explanatory questions (“Why is the tumor not resectable?”), which are essential to evaluating clinical reasoning and text generation. This is a major limitation, especially given that existing 3D VLMs like CT-CHAT and OmniMRI (https://arxiv.org/abs/2508.17524) explicitly benchmark such capabilities. Short, factoid-level questions cannot fully assess models’ real-world utility in generating clinically meaningful responses.

**Questions:**

In addition, I have the following questions:

1. How do state-of-the-art “generalist” VLMs such as Med-PaLM, MedGemini, and LLaVA-Med perform on CTLesionVQA tasks compared to the 3D models benchmarked here?
2. How is class imbalance handled during the training?
3. The authors explicitly state the dataset is “not for clinical deployment” and that current VLMs “are not yet ready for complex clinical diagnostics”. However, is this because there is no real radiology reports, involved in this dataset? What would happen if one were to evaluate some on a real-world volume-report dataset?

---

### Official Review · Reviewer_eQ59 · 2025-10-20

**Soundness:** 2
**Presentation:** 2
**Contribution:** 2
**Rating:** 2
**Confidence:** 3

**Summary:**

This work presents CTLesionVQA, a VQA benchmark for abdominal lesions in CT scans that combines data across several public datasets. Questions for VQA are generated by applying CLEVR to the CT lesion analysis domain. The authors evaluate several existing VLMs on this new benchmark to determine which problems are the most challenging.

**Strengths:**

- This work points out an important limitations of existing CT benchmarks.
- Interesting analysis on which how dataset and model characteristics  may impact downstream performance.

**Weaknesses:**

- **Unclear what dataset methodologies are new contributions**. Is CTLesionVQA a direct application of CLEVR to CT images or does it introduce new dataset principles as well? Since the CTLesionVQA is based on CLEVR framework, I would expect a more extensive discussion on what this framework is in related works.
- **Dataset Feature Clarity could be improved**: The first feature (line 150) of CTLesionVQA (high data volume and diversity) is vaguely defined. Can you clarify what constitutes high volume and diversity? Is this defined with respect to different populations, different diseases, different sensor types, different data collection procedures, etc.?
- **Some analysis seem to be incomplete**:
    - Why does section 4.3 and 4.4 consider only performance of RadFM? Shouldn’t we be comparing performance across all VLM models to find shared challenges between all models? Right now, its unclear if the hypothesis in section 4.3 is specific to RadFM or not.
    - The description of the oracle (line 431) is not clear. How do you know that the crops from nnUNet are reliable? Why does evaluation in 4.5 only consider nnM3D and not the other nnVLM models?
    - Most of the evaluations are centered around recognition performance (section 4.4, 4.5). I would recommend the authors provide an analysis of the medical and visual reasoning tasks as well to understand what kinds of questions are challenging for current methods. Are there any robust patterns that reveal insight into why reasoning questions are especially challenging?
- It’s not clear to me that some of the statements are reliable. For instance, the authors claim “vision architecture is critical” (line 319) since Merlin adopts 3D ResNet while other methods do not. However, this observation could be confounded by several other differences in model and training approaches (different LLMs, image size, projectors, etc.). To make sure that this is a robust observation, I would expect an experiment where replacing 3D ViT vision encoders are with 3D ResNets leads to consistent performance improvements across all FMs. Same comment about “Input Resolution shows weak correlation with performance” (line 323).

**Questions:**

- Are all question types equally distributed across different organs? Based on Fig 2, it seems like medical reasoning questions mostly consists of liver and pancreas cases, and no kidney or colon cases. Would it be possible to expect medical reasoning performance to generalize to organs where we do not have available VQA samples?
- How do we know that the generated questions are relevant to a standard clinical workflow? How can we make sure that solving VQA in the proposed benchmark will transfer to real world use cases?

---

### Official Review · Reviewer_KsYh · 2025-11-01

**Soundness:** 3
**Presentation:** 4
**Contribution:** 3
**Rating:** 6
**Confidence:** 2

**Summary:**

This paper presents CTLesionVQA, a large-scale benchmark to evaluate 3D medical Vision-Language Models (VLMs) on lesion-centric visual question answering tasks. The dataset contains 9,262 CT volumes from 17 public sources, with 395K expert-curated QAs spanning four diagnostic categories.
The benchmark is designed to assess whether VLMs can achieve the precision, reasoning, and domain understanding required for clinical diagnosis. Extensive evaluations on four advanced models (RadFM, M3D, Merlin, CT-CHAT) reveal that while VLMs handle measurement tasks reasonably well, they still underperform on lesion recognition and diagnostic reasoning. The study provides valuable insights into architectural and training factors affecting 3D VLM performance and introduces an organ-localization preprocessing strategy that boosts sensitivity for small lesions.

**Strengths:**

The paper tackles the readiness of multimodal models for real 3D clinical diagnosis through a well-designed and comprehensive benchmark. CTLesionVQA features high-quality expert annotations, clinically meaningful question templates, and well-structured functional programs to generate verifiable QAs. The dataset’s scale and diversity, aggregated from 88 centers, substantially exceed prior benchmarks, and its hierarchical design effectively mirrors real diagnostic workflows. The analysis is deep and thoughtful, identifying key limitations in existing 3D VLMs and demonstrating the role of architectural choices, fine-tuning, and anatomical priors in model performance. The proposed nnUNet-based preprocessing further strengthens the practical relevance by showing that anatomical localization meaningfully enhances recognition.

**Weaknesses:**

The primary limitation lies in the scope of evaluated models and interpretability depth. Although four strong baselines are included, they represent early-generation 3D VLMs. The paper would benefit from comparisons with more general multimodal LLMs (e.g., MedTrinity-25M) to better situate the benchmark in the broader MLLM landscape. Furthermore, while the benchmark covers diverse clinical question types, its reasoning dimension is limited to relatively simple 1–2 step tasks. The study could discuss how CTLesionVQA might evolve to probe higher-order reasoning or temporal integration.

**Questions:**

See above.

---

### Meta-Review · Area_Chair_d68p · 2026-01-01

**Summary:**

Although VLM produces results for 2D visual tasks, its application to 3D clinical diagnoses, which require recognition accuracy, reasoning ability, and medical knowledge, remains unclear. Therefore, this paper proposes the diagnostic VQA benchmark, "CTLesionVQA," for abdominal CT lesions. The authors compiled 9,262 CT volumes (approximately 3.7 million slices) and 395,000 expert questions from 17 public datasets. They then categorized these questions into four groups: recognition, measurement, visual reasoning, and medical reasoning. Four state-of-the-art VLMs were evaluated, and while their performance on measurement tasks was consistent, their performance on lesion recognition and reasoning tasks was insufficient and failed to meet clinical requirements. Furthermore, the evaluation demonstrated the importance of large-scale multimodal pretraining and showed that image preprocessing and vision module design significantly impact 3D perception.

Reviewers raised several concerns. For instance, comparisons were limited to early-generation, four-model architectures, which left its positioning relative to more general-purpose MLLMs unclear. The novelty of the dataset, the explanation of the CLEVR framework, and the definition of features were unclear. There were problems with unfairness in pre-training differences, simplification of inference design, insufficient QA quality verification, and the absence of lesion subtypes and multi-stage explanatory inference.

The authors did not address any of these concerns, leaving them unresolved.
Therefore, the AC has decided to reject this paper.

**Reviewer Concerns:**

Because the authors did not submit a rebuttal, all concerns remain unresolved.

**Reviewer Scores:**

This paper received negative evaluations from all reviewers. Since the authors did not submit a rebuttal addressing the reviewers’ concerns, the scores did not change.

---

### Decision · Program_Chairs · 2026-01-26

Reject